# An $O_2$-sensing stressosome from a Gram-negative bacterium

Xin Jia[1,*], Jian-bo Wang[1,*], Shannon Rivera[1], Duc Duong[2] & Emily E. Weinert[1]

Bacteria have evolved numerous pathways to sense and respond to changing environmental conditions, including, within Gram-positive bacteria, the stressosome complex that regulates transcription of general stress response genes. However, the signalling molecules recognized by Gram-positive stressosomes have yet to be identified, hindering our understanding of the signal transduction mechanism within the complex. Furthermore, an analogous pathway has yet to be described in Gram-negative bacteria. Here we characterize a putative stressosome from the Gram-negative bacterium *Vibrio brasiliensis*. The sensor protein RsbR binds haem and exhibits ligand-dependent control of the stressosome complex activity. Oxygen binding to the haem decreases activity, while ferrous RsbR results in increased activity, suggesting that the *V. brasiliensis* stressosome may be activated when the bacterium enters anaerobic growth conditions. The findings provide a model system for investigating ligand-dependent signalling within stressosome complexes, as well as insights into potential pathways controlled by oxygen-dependent signalling within *Vibrio* species.

[1] Department of Chemistry, Emory University, 1515 Dickey Drive, Atlanta, Georgia 30322, USA. [2] Integrated Proteomics Core, Emory University, 615 Michael Street, Atlanta, Georgia 30322, USA. * These authors contributed equally to this work. Correspondence and requests for materials should be addressed to E.E.W. (email: emily.weinert@emory.edu).

Bacteria of the *Vibrio* genus are rod-shaped, Gram-negative gamma-proteobacteria that are highly abundant in marine waters and include a number of pathogenic species, such as *Vibrio cholerae*, which causes acute diarrhoea, and *Vibrio vulnificus*, which causes life-threatening skin and soft tissue infections and has an ~50% mortality rate[1,2]. Pathogenic *Vibrio* species have evolved to survive in both marine environments and within their aquatic/mammalian hosts, necessitating changes in gene expression as the bacteria pass between these disparate environments[3,4]. In addition, *Vibrio* species alter global phenotypes, such as biofilm formation, capsule production and virulence factor production, in response to decreased $O_2$ levels[5–8]. A number of gene clusters that control expression of genes directly or indirectly linked to bacterial virulence have been characterized; however, additional pathways that regulate gene expression levels in response to stress adaptation and bacterial pathogenesis of the *Vibrio* genus have yet to be identified[9,10]. Within Gram-positive bacteria, such as *Bacillus subtilis*, the stressosome, an ~1.8 MDa signalling complex, alters gene expression levels through activation of $\sigma^B$ in response to diverse environmental stresses such as temperature, pH and energy source depletion[11,12]. Recently, putative homologues encoding core stressosome components (*rsbR*, *rsbS* and *rsbT*) were identified in the genomes of various *Vibrio* species[13] and, within *V. vulnificus*, *rsbR/S/T* transcript levels were found to be upregulated on growth in sea water[14], suggesting that stressosome signalling within *Vibrio* species is important for sensing and survival in estuarine environments. While a stressosome signalling complex from Gram-negative bacteria has yet to be characterized, these results suggest that a stressosome-like pathway may be involved in stress responses in *Vibrio* species.

Within Gram-positive bacteria, the core signalling complex, formed by RsbR, RsbS and RsbT, senses and transmits the stress or signal to $\sigma^B$ through activation of downstream proteins (Fig. 1)[15,16]. However, unlike paradigmatic two component signalling pathways[17], the stressome pathway in *B. subtilis* activates a cascade of proteins, including phosphatases and additional kinases, that results in release of $\sigma^B$ from an anti-$\sigma$ factor[15,16,18,19]. The *rsbR* homologues encode a variable N-terminal sensing domain, and a C-terminal STAS (sulfate transport and anti-anti-$\sigma$ factor) domain[18,20,21]. Within *Bacillus* species, RsbR proteins contain non-haem globin domains, which are domains that maintain a globin fold but have introduced mutations that prohibit haem binding[21]. The *B. anthracis* RsbR non-haem globin domain was crystallized bound to a fatty acid; however, the *in vivo* signalling molecules sensed by RsbR-containing stressosomes remain unknown. RsbS consists of a STAS domain that makes protein–protein interactions with RsbR to form the core of the stressosome complex, and RsbT is a kinase that transmits the stress signal initially via phosphorylation of serine/threonine residues in RsbR and RsbS, and then through phosphorylation of downstream proteins, ultimately resulting in transcriptional changes of over 150 genes that are involved in stress response[11,12,16,18,20,22–24]. However, elucidating the mechanism of stress sensing and signal transduction within stressosomes has been challenging, as the activating signals for RsbR proteins within Gram-positive bacteria remain elusive, and putative stressosome complexes within Gram-negative bacteria remain uncharacterized.

Putative RsbR homologues from *Vibrio* species comprise a unique class of stressosome sensor proteins in which the N-terminal domain consists of a haem-binding sensor globin domain, rather than the non-haem globins found in *Bacillus* species. On the basis of homology with other globin coupled sensors, the N-terminal domain of RsbR in *Vibrio brasiliensis* maintains the proximal histidine residue required for haem binding, and the distal tyrosine involved in stabilization of bound ligands are conserved[25–27]. In contrast, those key residues are not conserved in previously characterized RsbR proteins from *B. subtilis* and other Gram-positive bacteria[25]. As previous studies have demonstrated, members of the *Vibrio* genus respond to anaerobiosis by promoting virulence gene induction, triggering biofilm dispersal and decreasing capsule production[5–8]. While some of the pathways regulating these phenotypes are known[5,6], $O_2$-dependent stressosome signalling likely plays key roles in *Vibrio* species in adapting to changing $O_2$ levels in the marine environment, and also may control virulence of *Vibrio* pathogens as they transition to the anaerobic/microaerobic gut environment[5–8,28–30]. Recent work on the stressosome from *Moorella thermoacetica* found that the stressosome also regulates cyclic di-GMP signalling through phosphorylation of a down-stream diguanylate cyclase by the RsbT kinase homologue[31]. As cyclic di-GMP is a major regulator of biofilm formation, these data provide further evidence that the stressosome pathways within *Vibrio* species may directly regulate $O_2$-dependent biofilm formation and capsule production[31]. Furthermore, characterization of an $O_2$-sensing RsbR protein and stressosome complex will allow for detailed analysis of the mechanism of ligand-dependent signal transduction between RsbR proteins and the stressosome complex, as well as identification of stressosome-responsive genes that are controlled in response to changes in environmental $O_2$ levels.

Here we report the first example, to our knowledge, of a functional stressosome from a Gram-negative bacterium that senses $O_2$ through a haem-bound RsbR.

## Results

**Characterization of RsbT kinase activity.** To characterize the putative *V. brasiliensis* stressosome, candidate *rsb* genes encoding homologues of the stressosome components RsbR, RsbS and RsbT in *V. brasiliensis* were cloned into expression vectors and heterologously expressed (Supplementary Fig. 2). On purification, RsbR was found to bind haem, as predicted, and exhibited spectra characteristic of a histidyl-ligated sensor globin (Fig. 2, Supplementary Fig. 1 and Supplementary Table 1)[32,33]. The

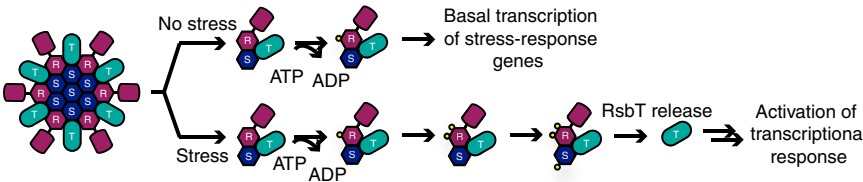

**Figure 1 | Model of the stressosome previously characterized in *B. subtilis*.** In the absence of stress, RsbT (turquoise) only phosphorylates RsbR (purple), resulting in basal transcription level (top). In the presence of an environmental stressor, RsbT continues to phosphorylate RsbR; increasing levels of RsbR phosphorylation causes RsbS (blue) to also be phosphorylated by RsbT. Phosphorylated RsbS reduces the binding affinity of RsbT to the stressosome core, causing RsbT to activate downstream stress response gene transcription through a cascade that activates $\sigma^B$ (bottom)[15,16].

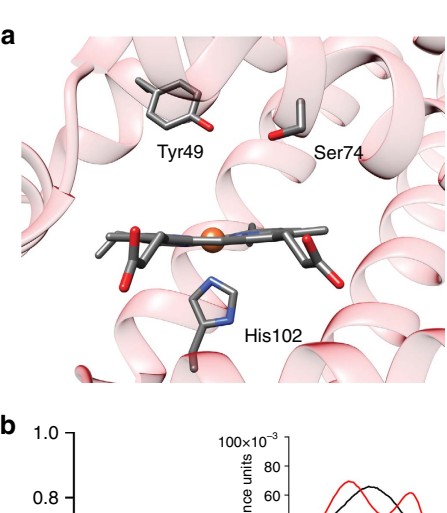

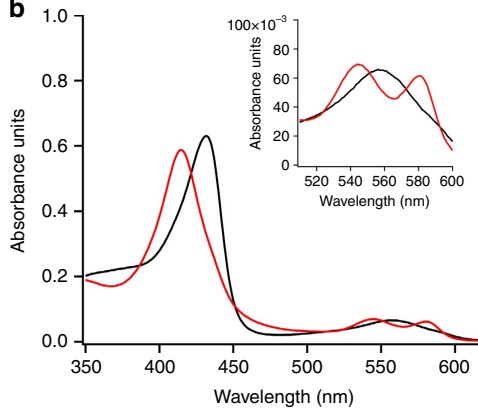

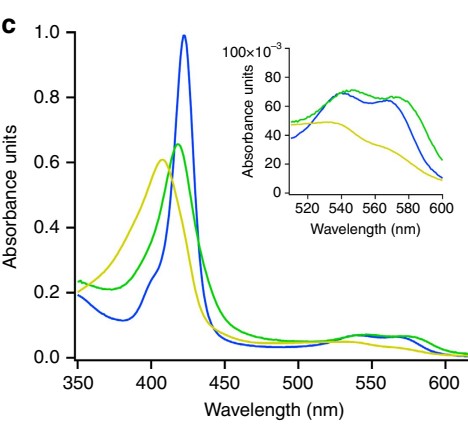

**Figure 2 | RsbR haem pocket model and haem spectra.** (**a**) Homology model of the haem pocket of *V. brasiliensis* RsbR generated using Phyre2 (based on PDBID: 1OR4)[57]. The haem, proximal histidine and distal pocket hydrogen bond donors (tyrosine and serine) are shown. (**b**,**c**) Ultraviolet–visible spectra of RsbR in *V. brasiliensis*. Expanded $\alpha/\beta$ band region is shown in the inset. Wavelength maxima are listed in Supplementary Table 2. (**b**) Black, $Fe^{II}$; red, $Fe^{II}$-$O_2$. (**c**) Blue, $Fe^{II}$-CO; green, $Fe^{II}$-NO; yellow, $Fe^{III}$. *V. brasiliensis* $Fe^{III}$ RsbR consists of a mixture of five- and six-coordinate haem species.

putative kinase RsbT was found to be active and exhibited time-dependent autophosphorylation (Supplementary Fig. 3). Previous studies on RsbT from *B. subtilis* identified an aspartate residue that is crucial for kinase activity[34]. Mutation of the homologous aspartate in *V. brasiliensis* RsbT (D87N) resulted in complete loss of kinase activity, suggesting that key active site residues responsible for stressosome-associated kinase activity are conserved (Supplementary Fig. 4A).

Within the stressosome complex, binding and controlled release of kinase RsbT are controlled through sensing of the stress signal by RsbR (Fig. 1)[23,35]. In the absence of stress, basal RsbT phosphorylation activity results in low levels of phosphorylated RsbR, which is a prerequisite for subsequent signal transduction in response to stress[20,23]. After activation, an increasing level of phosphorylated RsbR facilitates phosphorylation of RsbS by RsbT within the stressosome, decreasing affinity of RsbT for the RsbR/S stressosome core[11,22,23]. To test the role of RsbR ligand sensing in controlling the phosphorylation cascade, effects of ligand binding to the sensor globin domain of RsbR on RsbT kinase activity were measured within the binary mixture. The rates of phosphorylation of RsbR by RsbT varied depending on the ligation state of RsbR, with $Fe^{II}$ RsbR leading to $\sim$3.5-fold faster phosphorylation rates than $Fe^{II}$-$O_2$ RsbR (Supplementary Fig. 5). Furthermore, time-dependent phosphorylation of RsbS by RsbT, in the absence of RsbR, is not observed for the *V. brasiliensis* system (Supplementary Fig. 4C). Despite the lack of phosphoryl transfer, RsbS and RsbT are able to associate in the absence of RsbR (Supplementary Fig. 6). These data show that similar to the *B. subtilis* stressosome RsbR is required for the subsequent phosphorylation of RsbS and further signal transduction[11,20,23], highlighting the importance of the sensor protein RsbR in signal transmission within the stressosome core.

**Ligand-dependent activity of *V. brasiliensis* stressosome.** To determine if RsbR can serve as a haem-based sensor within fully reconstituted stressosome core complex and to determine potential physiologically relevant ligands, kinase activity of RsbT in RsbR/S/T mixtures was investigated with RsbR in various ligation and oxidation states (Fig. 3 and Supplementary Fig. 7). Kinase RsbT was found to be active when RsbR is in the $Fe^{II}$ unliganded state, similar to the RsbR/T binary mixture. RsbR $Fe^{II}$-NO and $Fe^{II}$-CO yielded intermediate rates of phosphoryl transfer by RsbT to RsbR and RsbS, while $Fe^{II}$-$O_2$ and $Fe^{III}$ resulted in basal kinase activity (Fig. 3, Supplementary Fig. 7 and Supplementary Table 2). Similar effects of ligation/oxidation state have been observed for other members of the globin-coupled sensor family with different output domains, potentially due to either strong interactions between bound $O_2$ with distal pocket tyrosine and serine residues or conformational rearrangements associated with altered haem electronics due to the change in oxidation state[25,26,33,36–44]. While the observed rates and levels of RsbT autophosphorylation are constant (Fig. 3c) for all ligation/oxidation states, the overall kinase activity (autophosphorylation and transfer to RsbR/RsbS) is clearly increased for RsbR $Fe^{II}$. Therefore, levels of phosphorylated RsbT likely remain similar due increased rates of transfer to RsbR and RsbS. These results suggest that ligand-dependent regulation of RsbT activity may be caused by a conformational change that limits RsbT activity on binding of molecular $O_2$ to the haem of RsbR and/or interactions between $Fe^{II}$ RsbR and RsbS/T that lead to activation of RsbT activity.

In contrast, phosphorylation of RsbS in the RsbR/S/T mixture was undetectable when RsbR was included in the $Fe^{II}$-$O_2$ or $Fe^{III}$ states, even though inclusion of RsbR is required for RsbS phosphorylation (Supplementary Fig. 4C). RsbR and RsbS, in the absence of RsbT, are able to associate regardless of RsbR ligation state, as are all three proteins of the stressosome complex (Supplementary Fig. 6). Therefore, changes in RsbR ligation state do not affect formation of the STAS domain-mediated core complex between RsbR and RsbS, which could potentially affect RsbT activation/recruitment. Instead, these data suggest that RsbR/RsbS interactions are crucial for controlling RsbT phosphorylation of RsbS, either by inhibiting RsbS phosphorylation by RsbT in the presence of $Fe^{II}$-$O_2$ or $Fe^{III}$ RsbR, possibly by limiting

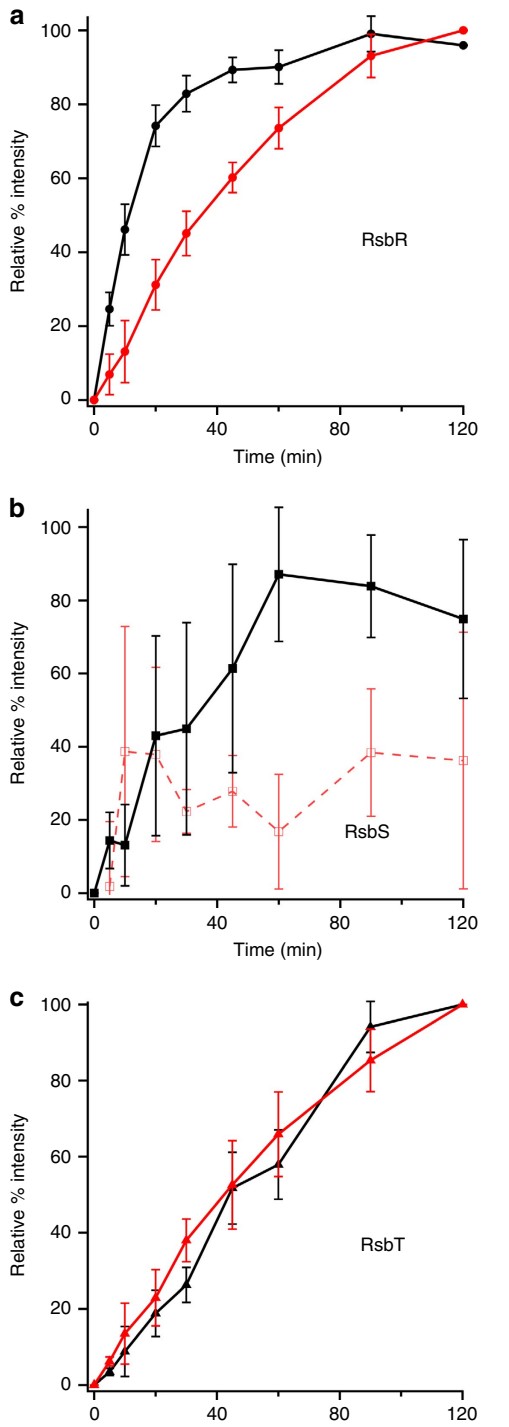

**Figure 3 | Phosphorylation kinetics of kinase RsbT in the presence of RsbR in various ligation states and RsbS.** Average rate of phosphorylation of RsbR (**a**) RsbS (**b**) and RsbT (**c**) in a RsbR/S/T mixture (ligation/oxidation states of RsbR: $Fe^{II}$-$O_2$, red; $Fe^{II}$ unligated, black). All experiments were performed at least in triplicate, and mean and s.d. values are plotted for each time point. Phosphorylation of RsbS was not detected above background levels when $Fe^{II}$-$O_2$ RsbR was included in the reaction mixture and therefore did not show any time-dependent kinetics (variability in the RsbS $Fe^{II}$-$O_2$ plots is due to normalization within the background signals).

RsbT access to the RsbS phosphorylation site, or through generation of a productive phosphoryl-transfer complex in the presence of $Fe^{II}$ RsbR. While inclusion of $Fe^{III}$ RsbR results in

lack of transfer to RsbS and slow rates of transfer to RsbR, in the RsbR/S/T mixture, $Fe^{II}$-$O_2$ RsbR is stable over the course of the reactions and auto-oxidizes at a rate of $0.24\,h^{-1}$, similar to myoglobin (Supplementary Fig. 8)[44]. Therefore, it is unlikely that RsbR is serving as a redox sensor *in vivo*, and RsbR most likely controls $O_2$-dependent stressosome signalling.

**Identification of phosphorylation sites.** As Rsb proteins from *V. brasiliensis* share low amino-acid sequence identity with those from *B. subtilis* (RsbR = 26%, RsbS = 28% and RsbT = 18%), a number of previously identified phosphorylation sites are not conserved[18,24,35,45]. Therefore, a mass spectrometry (MS)-based approach was used to identify phosphorylation sites on *V. brasiliensis* Rsb proteins. Analysis of a RsbR/S/T phosphorylation assay identified thiophosphorylated peptides of RsbR and RsbT that were observed in the kinase reaction, but not the control reaction (Supplementary Table 3 and Supplementary Fig. 9). Sites of RsbR phosphorylation were observed at the end of a conserved linker region between the sensor globin and STAS domains (*Vb* RsbR S178 and T182). Previous work on *B. subtilis* RsbR observed that mutations to the linker region α-helix led to altered levels of basal stressosome activity[46], suggesting that phosphorylation of this region within *Vb* RsbR may lead to similar effects *in vivo*. Unfortunately, no thiophosphorylated peptides from RsbS were identified; this likely is due to the lower levels of RsbS thiophosphorylation observed, as compared with RsbR and RsbT (Supplementary Fig. 4B). As the sites of phosphorylation previously identified on *B. subtilis* RsbS are not conserved in *V. brasiliensis* RsbS, future experiments will be required to identify the phosphorylation site(s).

Phosphorylation sites on kinase RsbT were identified in an α-helix that is predicted to provide key interactions for RsbT dimerization and for σ factor binding, and is close to anti-anti-σ factor SpoIIAA-binding site, based on co-crystal structures of *Bacillus stearothermophilus* SpoIIAB (an anti-σ factor and homologue of RsbT) with $\sigma^F$ and SpoIIAA (refs 47,48). Previous work on SpoIIAB-type kinases from various *Bacillus* species did not identify sites of phosphorylation on the kinases or detect auto-phosphorylated species, potentially due to the low sequence identity in the phosphorylated region of *Vb* RsbT and/ or the resulting labile phosphorylated amino-acid products (rather than the more stable thiophosphorylated products observed for *Vb* RsbT). The highest intensity peptide identified is phosphorylated at S39 (*Vb* RsbT numbering; Supplementary Table 4 and Supplementary Fig. 9); the homologous position directly interacts with $\sigma^F$ in the *B. stearothermophilus* RsbT-$\sigma^F$ co-crystal structure[47]. Given that Gram-negative bacteria do not have the conserved σ factors found in Gram-positive *Bacillus* species, the downstream binding partner for *Vb* RsbT is unknown. However, these data suggest that altering the electrostatics of this face of RsbT leads to stressosome activation, possibly by altering protein–protein interactions. The identification of phosphorylation sites on *V. brasiliensis* proteins can be used to drive future investigations to determine *in vivo* phosphorylation patterns and determine if different levels of environmental stress (varying $O_2$ tensions for the *Vb* stressosome) affects the extent of phosphorylation, as was observed for the *B. subtilis* stressosome[24].

**Stressosome particle size.** Dynamic light scattering (DLS) experiments of individual proteins and RsbR/S/T mixture were performed to provide independent validation of the protein–protein interactions and to determine the approximate size of the *V. brasiliensis* stressosome complex (Supplementary Fig. 10). A previously reported cryo-EM structure of the *B. subtilus*

stressosome core reported a diameter of ∼30 nm (ref. 11). DLS of *V. brasiliensis* proteins shows that RsbR and RsbS form large oligomers (15.1 and 20.5 nm, respectively), likely interacting through STAS domain interactions, while the kinase RsbT forms considerably smaller oligomers (diameter = 9.8 nm). In contrast, the RsbR/S/T mixture forms a complex with diameter of ∼16 nm, which is smaller than the oligomers formed by individual RsbR and RsbS proteins, demonstrating that hetero-oligomeric protein–protein interactions between three proteins, likely mediated by RsbR and RsbS STAS domains, are different than interactions between homo-oligomers. A similar trend is observed for the individual proteins and RsbR/S/T complex following cleavage of the solubility tags, with RsbR and RsbS forming very large oligomers (34 and 43.3 nm, respectively), RsbT forming smaller oligomers (27.3 nm), and the RsbR/S/T complex forming large oligomers with a diameter of 30.2 nm (Supplementary Figs 10b and 11). The large particles observed for the individual proteins suggests that cleavage of the solubility tag decreases stability of the proteins, leading to association/aggregation. However, formation of the RsbR/S/T complex results in particle diameters nearly identical to those observed for the *B. subtilis* stressosome, suggesting that *V. brasiliensis* RsbR/S/T proteins can associate into a stressosome-like complex. Ongoing work is focused on generating additional *V. brasiliensis* RsbR/S/T expression constructs and elucidation of the complex structure using cryo-EM. Regardless, DLS measurements mirror the trend in RsbT phosphorylation kinetics and $O_2$ dissociation rates, highlighting that the three-protein complex RsbR/S/T forms key interactions involved in *V. brasiliensis* stressosome signal transduction.

**Oxygen-binding kinetics.** The response of the *V. brasiliensis* stressosome to ligand binding at the haem of RsbR provides a unique opportunity to interrogate the effects of RsbR/S/T protein interactions on signal sensing by RsbR. For haem proteins, $O_2$ association and dissociation rates are very sensitive and can report on the state of the haem pocket, including changes in hydrogen-bonding patterns/conformation, electronics and access to the pocket[25,44,49,50]. Therefore, RsbR $O_2$-binding kinetics were determined to quantify changes in the RsbR-sensing domain in the absence and presence of RsbS and RsbT (Table 1). $O_2$ binding to the haem within RsbR is not greatly affected by formation of a complex with RsbS/RsbT, with $O_2$ association rates of 5.2 and $3.6\,\mu M^{-1}s^{-1}$ for RsbR and RsbR/S/T, respectively. The $O_2$ dissociation rate of RsbR in *V. brasiliensis* exhibits bi-exponential

kinetics, suggesting that RsbR has multiple conformations of the haem pocket (Supplementary Fig. 12), as has been observed for other globin-coupled sensors (Table 1)[36,37,50]. On the basis of sequence alignments, distal pocket tyrosine and serine residues are predicted to provide key hydrogen-bonding residues in the haem distal pocket and interact with bound ligands (Supplementary Fig. 1 and Fig. 2a). Although the large complexes of RsbR complicate detailed analysis, the biphasic $O_2$ dissociation kinetics suggest that multiple conformations of the haem pocket, including hydrogen-bonding patterns of these amino acids with bound $O_2$, yield multiple dissociation rates[36,39]. Formation of the RsbR/S/T complex alters $O_2$ dissociation from RsbR; both the slow ($k_1$) and fast ($k_2$) rates of individual protein RsbR and RsbR within RsbR/S/T complexes remain unchanged, but percentages of each rate are altered. These results suggest that RsbR/S/T interactions result in a greater percentage of RsbR haem pockets being in a high-affinity conformation (7.5% versus 19.6%; Table 1). Thus, formation of RsbR/S/T complexes not only alters phosphoryl transfer rates from RsbT and allows for transfer to RsbS but also likely changes the conformation of sensor protein RsbR, altering the effective $O_2$ affinity. Given that the calculated $O_2$ affinities are ∼2 and ∼15 μM, *V. brasiliensis* stressosome will be able to sense a range of $O_2$ levels as the bacteria travel through the water column. By having two, distinct $O_2$ affinities, the organism should be able to turn on the stressosome signalling pathway as the $O_2$ levels fall below ∼10% of air-saturated levels, and then further activate the remaining RsbR sensors as the $O_2$ levels continue to drop into low micromolar concentrations. This two phase $O_2$ sensing may allow *V. brasiliensis* to modulate activation of downstream proteins in response to a wider range of $O_2$ concentrations.

**Prevalence of putative haem-bound RsbR proteins.** A phylogentic analysis[51] of sensor globin- and non-haem globin-containing RsbR proteins from representative species was performed and found two major clades, one containing *B. subtilis* RsbR and the other containing *V. brasiliensis* RsbR (Supplementary Fig. 13). Within the *V. brasiliensis* clade, all of the putative RsbR protein sensor globin domains contain the conserved proximal histidine and distal tyrosine residues (Supplementary Fig. 14)[25,52], suggesting that these proteins will be haem-bound and will reversibly bind $O_2$. In contrast, members of the *Bacillus* clade do not contain the residues required for haem and $O_2$ binding, suggesting that they function as non-haem globins. Furthermore, as a number of the bacteria within the

**Table 1 | O$_2$-binding kinetics.**

| Protein | $k_{on}$ (μM$^{-1}$s$^{-1}$) | $k_{1,\,off}$ (s$^{-1}$) | $k_{2,\,off}$ (s$^{-1}$) | %$k_1$ | %$k_2$ | $K_d$ (μM) |
|---|---|---|---|---|---|---|
| HemAT-*Bs*[*] | 19 | 87 | 1,900 | NR | NR | 4.6 |
| | | | | | | 100 |
| *Ec*DosC[†] | 0.9 | 13 | — | — | — | 14 |
| *Bpe*GReg | 7.0[‡] | 0.82[§] | 6.30[§] | 39.3[§] | 60.7[§] | 0.12 |
| | | | | | | 0.9 |
| *Pcc*GCS | 7.2 ± 0.3 | 0.56[§] | 3.87[§] | 56.1[§] | 44.0[§] | 0.08 |
| | | | | | | 0.54 |
| RsbR | 5.2 ± 0.1 | 7.0 ± 0.3 | 67.4 ± 0.8 | 7.5 ± 1.7 | 92.5 ± 1.7 | 1.4 |
| | | | | | | 13.0 |
| RsbR in RsbR/S/T complex | 3.6 ± 0.2 | 7.9 ± 0.1 | 65.0 ± 0.7 | 19.6 ± 1.4 | 80.5 ± 1.4 | 2.2 |
| | | | | | | 18.1 |

NR, not reported.
[*]Ref. 50.
[†]Ref. 26.
[‡]Ref. 38.
[§]Ref. 39.

*Vb*RsbR branch also are aquatic organisms, haem-containing RsbR proteins and $O_2$-dependent stressosome signalling potentially are broadly important for survival in changing $O_2$ concentrations, either within the aqueous environment or during the pathogenic-planktonic transition.

## Discussion

Stressosome-dependent signalling within *Vibrio* species likely involves the putative downstream phosphatase RsbU, which is a downstream protein in *B. subtilis* stressosome signalling and is localized in the *V. brasiliensis* genome with RsbR, RsbS and RsbT. However, other downstream proteins in *B. subtilis* stressosome cascade, including the anti-σ factor RsbV and kinase RsbW, are not found in the *V. brasiliensis* genome. Furthermore, Gram-negative bacteria, including *Vibrio* species, do not have σ[B], the protein required for initiating transcription of general stress response genes in *B. subtilis*[53]. Therefore, it is likely that the *V. brasiliensis* stressosome is controlling other downstream pathways, such as aerotaxis, due to the presence of a putative CheY-like protein next to the RsbR/S/T operon (Supplementary Fig. 15)[11,14]. Further studies investigating the downstream proteins and genes controlled by the *V. brasiliensis* stressosome will help to elucidate stressosome signalling pathways in Gram-negative bacteria.

In summary, we report the first example of a haem-bound RsbR as the sensing protein in a functional stressosome signalling complex, as well as the first characterization of a stressosome from a Gram-negative bacterium. These studies demonstrate a novel $O_2$-dependent stressosome signalling pathway that will be activated in $O_2$-depleted environments, and inactive once the organism enters $O_2$-rich environments. Our work provides a platform to study conformational changes within the stressosome complex on ligand binding, which will aid in dissecting key protein–protein and domain–domain interactions involved in activation of stress-responsive genes. Future *in vivo* work will allow for identification of ligand-dependent gene transcription and downstream effectors of the stressosome signalling pathway. Furthermore, understanding stressosome-regulated pathways in *V. brasiliensis* will help gain insights into stressosome-controlled signalling events in Gram-negative bacteria and the roles of $O_2$-dependent signalling in environmental sensing and pathogenicity of the *Vibrio* genus.

## Methods

**Cloning and site-directed mutagenesis.** The wild-type *rsbR*, *rsbS* and *rsbT* genes used in this study were amplified from genomic DNA of *V. brasiliensis* (DSMZ 17184). The primer sequences used in the overlap PCR reactions are listed in Supplementary Table 4. All oligonucleotides were synthesized by Integrated DNA Technologies. Target genes *rsbR*, *rsbS* and *rsbT* were cloned into the pMAL-c2X vector (Novagen), while *rsbS* was cloned into pGEX-4T1 vector (Novagen) for improved protein expression and stability. For construction of the *rsbS* expression plasmid, the PCR product was digested with endonucleases EcoRI and XhoI, and inserted into pGEX-4T1 vector via EcoRI and XhoI restriction sites. To construct His-tags at the N terminus of the maltose-binding protein (MBP) tags on RsbR and RsbT, the PCR products were digested with NdeI and EcoRI, and the purified NdeI-MBP-EcoRI fragment was inserted into the pMAL-c2X vector via NdeI and EcoRI restriction sites. The resulting plasmid was named pHis-MBP. The *rsbR* or *rsbT* target gene was subsequently ligated into the HindIII and EcoRI sites on pHis-MBP. A Tev protease site also was introduced between MBP/GST (glutathione-S-transferase) tag and the target genes. The ligation mixture was transformed into *Escherichia coli* DH5α cells. Positive transformants of all constructs were selected on Luria–Bertani (LB) plates containing 100 µg ml$^{-1}$ ampicillin, and DNA sequences were confirmed by sequencing (Eurofins). All cloning primers are listed in Supplementary Table 1.

The wild-type pHis-MBP-RsbT plasmid served as a template for site-directed mutagenesis (mutagenic primers are listed in Supplementary Table 4). Positive transformants of all constructs were screened on LB plates containing 100 µg ml$^{-1}$ ampicillin, and the DNA sequences were confirmed by sequencing.

**Protein expression and purification.** Expression plasmids for wild-type RsbR, RsbS, RsbT and mutant RsbT were transformed into chemically competent *E. coli* Tuner cells (Novagen). Cultures were grown in LB media at 37 °C until an optical density of 0.6–0.8 at 600 nm was reached. Cultures were then cooled to 18 °C and protein overexpression was induced by addition of 1 mM isopropyl-β-D-thiogalactoside and incubation with shaking 16–18 h. For RsbR expression, 500 µM 5-aminolevulinic acid was added before addition of 1 mM isopropyl-β-D-thiogalactoside to improve haem incorporation. All cells were collected by centrifugation and lysed by homogenization (Avestin) after resuspension in buffer containing 50 mM Tris-HCl, pH 7.0, 250 mM NaCl and 5% (v/v) glycerol. Soluble proteins were separated from cell debris by ultracentrifugation at 186,000*g* for 1 h. The supernatant with His$_6$-MBP-fused RsbR, RsbT and mutant RsbT were purified by nickel-NTA affinity chromatography. GST-tagged RsbS was purified by use of glutathione affinity chromatography. Purified proteins were desalted and further purified using a S200 gel filtration column (GE Healthcare) into storage buffer (50 mM tetraethylammonium, pH 7.5, 50 mM NaCl and 5% glycerol). Representative SDS–PAGE of purified samples is shown in Supplementary Fig. 2.

Tag-free RsbR, RsbS and RsbT were generated by incubating tagged proteins with TEV protease at 4 °C overnight. Cleaved His-MBP tags and TEV were removed using a Ni-NTA column, while GST tags were removed using a glutathione column. Tag-free proteins were collected in the flow through in storage buffer and concentrated before use.

**Electronic spectroscopy.** All spectra were recorded on an Agilent Cary 100 with Peltier accessory. Preparation of complexes was carried out as previously described[39,54] except that the proteins were prepared in storage buffer.

**Serine kinase assay using ATP-γ-S.** Autophosphorylation and phosphoryl transfer kinetic experiments using ATP-γ-S were performed as previously described[55]. Briefly, serine kinase RsbT (5 µM) auto-thiophosphorylation reactions were initiated by addition of 500 µM ATP-γ-S in reaction buffer containing 50 mM tetraethylammonium, pH 7.5, 50 mM NaCl, 10 mM MgCl$_2$ and 5% glycerol (buffer A). Thiophosphorylation reactions of RsbT (5 µM) in the presence of RsbR (5 µM) and RsbS (5 µM) in buffer A were initiated by the addition of 500 µM ATP-γ-S. The reactions were quenched with 6 µl of 500 mM EDTA at 0, 5, 10, 20, 30, 45, 60, 90 and 120 min. Quenched reactions were heated at 60 °C for 5 min, before alkylation with 1 mM *para*-nitrobenzylmesylate for 1.5 h. Proteins were separated on 4–20% Tris-glycine SDS–PAGE gels (Biorad) and then were transferred from SDS–PAGE gels to nitrocellulose transfer membranes (0.2 µm, BioRad) using the Trans-Blot Turbo Transfer System (BioRad) at the high-molecular-weight setting for 10 min. Thiophosphoserine was detected as previously described[55]. Primary antibody specific for alkylated thiophosphate ester (1:5,000, Abcam 51–8) was added and incubated with the blot overnight at 4 °C. Subsequently, secondary antibody goat anti-rabbit HRP (1:5,000, BioRad) was incubated with the blot at 25 °C for 1.5 h. The blot was developed using Western Blotting Detection Reagent kit (BioRad) and Amplified Opti-4CN Substrate kit (BioRad) for colorimetric detection. The blot was imaged using Epson Perfection V600 photo scanner (Epson) at the Professional Setting. Kinetics experiments were performed at least in triplicate to ensure reproducibility and accuracy of the assay, and mean and s.d. values were calculated for each time point. Negative control reactions were performed without addition of ATP-γ-S to ensure there was no cross-activity with alkylated cysteines. Representative western blots are shown in Supplementary Fig. 4.

While the *B. subtilis* stressosome previously was found to have a 2:1:1 RsbR:S:T stoichiometry[11], with RsbR proteins forming dimers in the full complex, changing the *V. brasiliensis* RsbR/S/T ratio, including 2:1:1, in initial phosphorylation tests did not alter the level of phosphorylation and time dependence of the reaction. Therefore, the 1:1:1 stoichiometry (described above) was used for all reactions because it resulted in the clearly detectable phosphorylation of all three proteins. Doubling the concentration of RsbR (to yield a 2:1:1 ratio) resulted in western blots where either the RsbR signal was outside the linear range at later time points or the RsbS signal was barely observable.

**Western blot analysis.** Intensities of western blot signals were analysed using ImageJ (RSB). Signal intensity at each time point was expressed as a relative percentage to the most intense band on the same western blot.

**Pull-down assays.** His-MBP tags were cleaved from RsbR and RsbT using TEV protease before analysis, as described above. His-MBP-RsbS, RsbR and RsbT (10 µM each) were mixed and then added to pre-washed Ni-NTA beads (20 µl), incubated for 10 min at room temperature, spun down, washed with 50 mM Tris-HCl, pH 7.0, 250 mM NaCl and 5% (v/v) glycerol, 25 mM imidazole, and eluted by heating in PAGE loading buffer. Control lanes included either untagged RsbR or RsbT and were treated as described above. Samples were analysed by denaturing PAGE.

**$O_2$ dissociation rate.** $O_2$ dissociation rates were measured as previously described[39]. RsbR was reduced with dithionite in an anaerobic chamber (Coy), desalted

and then mixed with $O_2$-saturated buffer or RsbS and RsbT in $O_2$-saturated buffer. RsbR and RsbR/S/T samples were rapidly mixed with a solution of sodium dithionite in storage buffer (final ditionite concentration = 5 mM; dithionite concentration of 0.5 mM also was tested and dithionite concentration was found to not affect the $O_2$ dissociation rate) in an SX20 stopped flow apparatus. The dissociation of $O_2$ from haem protein RsbR at 25 °C was monitored using the SX20 stopped flow equipped with a diode array detector and temperature controlled bath, and fit globally using Pro-KII software (Applied Photophysics). Additional fitting analysis was performed using Igor Pro (Wavemetrics).

**$O_2$ association rate.** Transient visible spectra were collected on a home-built nanosecond transient infrared system described elsewhere with minor modifications[54,56]. Briefly, a Nd:YAG (Continuum) 1,064 nm fundamental was frequency doubled in a harmonic generator (Quanta-Ray) with a BBO Crystal and the resulting 532 nm pump separated from the fundamental with a series of prisms. The pump power was adjusted to 65 μJ per pulse and the beam focused to a 620 μm diameter spot. The 405 nm robe (Thor Labs) was focused onto a silicon avalanche photodetector (Thor Labs). Transient spectra were collected at room temperature. To calculate $\Delta A$, the pre-photolysis signal was averaged and used as $I_0$, and pumped signal was used as $I$ ($\Delta A = I/I_0$). Transient visible spectra were analysed with IGOR Pro (Wavemetrics). Reported rates are the average of three rates from triplicate measurements, and the error reported as ± the s.d. of the three rates.

**RsbR autoxidation experiment.** Autoxidation experiments were carried out with a Cary 100 ultraviolet–visible spectrophotometer (Agilent Technologies) at 25 °C. Spectra of $Fe^{II}$-$O_2$ RsbR were taken every 15 min for 18 h, as previously described[54]. Haem ligands for $Fe^{II}$ and $Fe^{III}$ RsbR (CO and KCN, respectively, which have diagnostic spectra) were added after two hours to determine the oxidation state of RsbR.

**Stressosome complex structural analysis.** DLS experiments were carried out with a NanoPlus zeta/nano particle analyser (Particulate Systems) at 25 °C. The particle size distribution was derived from a deconvolution of the measured intensity autocorrelation function of the sample by the general-purpose mode algorithm included in the software. DLS measurements were performed in buffer containing 50 mM Tris-HCl and 250 mM NaCl (pH = 7) with 5% (v/v) glycerol. All experiments were performed in triplicate to ensure reproducibility and accuracy of the assay.

**Identification of phosphorylation sites.** RsbR/S/T kinase activity and control reactions were performed as described above. The kinase reaction was allowed to proceed for 2 h before being quenched to ensure high levels of thiophosphorylation. Following quenching with EDTA, samples were mixed with SDS–PAGE loading buffer and separated, as described above. Relevant molecular-weight gel bands were excised and diced into ∼1 mm cubes, destained with 50% acetonitrile (ACN) in 50 mM ammonium bicarbonate (ABC), dehydrated with ACN and vacuum dried down using a Savant speedvac (Thermo). Trypsin was added at a concentration of 10 ng μl$^{-1}$ (enough to fully hydrate the gel pieces) and samples were placed on ice for 30 min. The gel cubes were then covered with ABC buffer and digestion was allowed to proceed overnight. Peptides were extracted twice with 5% formic acid in 50/50 ABC/ACN solution. Each extraction consisted of 10 min of low vortexing in extraction buffer and three cycles of centrifugation with 1 min on and 1 min off per cycle. A final step of 100% ACN was used to extract dehydrate the gel pieces and the entire peptide solutions were completely dried down using a Savant speedvac (Thermo).

Derived peptides were resuspended in peptide 10 μl of loading buffer (0.1% formic acid, 0.03% trifluoroacetic acid and 1% acetonitrile). Peptide mixtures (2 μl) were separated on a self-packed C18 (1.9 um Dr Maisch, Germany) fused silica column (20 cm × 75 uM internal diameter; New Objective, Woburn, MA) by a Dionex Ultimate 3000 RSLCNano and monitored on a Fusion mass spectrometer (ThermoFisher Scientific, San Jose, CA). Elution was performed over a 140 min gradient at a rate of 300 nl min$^{-1}$ with buffer B ranging from 3 to 80% (buffer A: 0.1% formic acid in water; buffer B: 0.1% formic in acetonitrile). The mass spectrometer cycle was programmed to collect at the top speed for 3 s cycles. The MS scans (400–1,500 $m/z$ range, 200,000 AGC, 50 ms maximum ion time) were collected at a resolution of 120,000 at $m/z$ 200 in profile mode, and the HCD MS/MS spectra (2 $m/z$ isolation width, 30% collision energy, 10,000 AGC target, 35 ms maximum ion time) were detected in the ion trap. Dynamic exclusion was set to exclude previous sequenced precursor ions for 20 s within a 10 p.p.m. window. Precursor ions with +1, and +8 or higher charge states were excluded from sequencing.

Spectra were searched using Proteome Discoverer 2.1 against *E. coli* RefSeq database (version 54 with 4,195 target sequences including the sequences for RsbR, RsbS and RsbT). Searching parameters included fully tryptic restriction and a parent ion mass tolerance (± 10 p.p.m.). Methionine oxidation (+ 15.99492 Da), asparagine and glutamine deamidation (+ 0.98402 Da), and serine, threonine and tyrosine thiophospate (+ 95.94349 Da) were variable modifications (up to 5 allowed per peptide). Percolator was used to filter the peptide spectrum matches to a false discovery rate of 1%. RsbR and RsbT has 93% sequence coverage, while RsbS

has 81% sequence coverage. A listing of identified thiophosphorylated peptides can be found in Supplementary Table 4.

**Data availability.** The data that support the findings of this study are available from the corresponding author on request.

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

## Acknowledgements

This work was supported by US National Science Foundation grant CHE 1352040 (E.E.W.) and a University Research Committee Grant from Emory University (E.E.W.). We thank B. Chica, G.E. Vansuch and Prof. R.B. Dyer for assistance with oxygen association rate and DLS measurements, and members of the Weinert group for helpful comments.

## Author contributions

X.J., J.W. and E.E.W. designed the study; X.J., J.W., S.R., D.D. and E.E.W. performed all experiments; X.J., J.W., D.D. and E.E.W. analysed the data; X.J. and E.E.W. wrote the manuscript.

## Additional information

**Competing financial interests:** The authors declare no competing financial interests.

