## [Peer Review File · Nature Communications]

Reviewers' comments:

Reviewer #1 (Remarks to the Author):

Summary of Results

Jia and colleagues present the characterisation of a new oxygen-sensing stressosome from a Gram negative bacterium *Vibrio brasiliensis*. The authors produce recombinant RsbR/S/T stressosome proteins from this species and characterise the protein complexes, kinase activity of RsbT and activity of the heme-containing RsbR protein.

Novelty

This is an interesting and novel study on a new class of stressosome that identifies the physiological signal for stressosome activation.

Data and methodology

Figure 1 is a very clear depiction of the stressosome, although in the case of *Bacillus subtilis* there is no evidence that RsbT is phosphorylated. There is no published experimental evidence to suggest that the RsbT/SpoIIAB type protein kinases are autophosphorylated.

The authors produce the RsbR/S/T proteins as fusions with MBP/GST and purify these by affinity chromatography. The use of fusion proteins is justified as a way of enhancing protein stability, although this can affect protein/protein interactions in protein complexes.

The heme-containing RsbR protein is characterised by UV/visible spectroscopy and the authors identify absorption peaks consistent with the presence of heme in the protein. The spectrum shown in Figure 2b lacks any indication of values, or units on the y-axis. I appreciate that for UV/visible spectroscopy these units are arbitrary, but it would be instructive to see the absolute values recorded.

The phosphorylation kinetics experiments presented suggest that RsbT is competent to phosphorylate RsbR and RsbT in the presence of RsbR in various ligation states. These results do not show particularly strong effects for different ligation states in the case of RsbR, while ligation with oxygen and the FeIII states appear to suppress RsbS phosphorylation. Given the size of the errors on these data it is not clear that the effects of the different ligation states of RsbR and RsbT are significantly different.

I have significant concerns about the identification of autophosphorylation of RsbT. There are no reports of this type of kinase exhibiting such behaviour and this observation requires strong corroborative evidence to rule out potential false positives. It would be instructive to see a non-denaturing PAGE gel with RsbT incubated with ATP to see if this protein is autophosphorylated would provide strong evidence to show that this effect is indeed real. Mass spectrometry could also be used to identify the position on which RsbT is autophosphorylated.

The Dynamic light scattering experiments used to assess the formation of stressosome complexes shown in Figure S7 appear to show that RsbR and RsbS form large aggregated of around 30 nm

diameter and the R/S/T mixture is much smaller at around 12 nm. These results give rise to serious concerns about the state of these proteins in solution. It would be highly instructive to attempt gel-filtration of the individual proteins and protein mixtures to assess whether the proteins are indeed forming aggregates. Simple negative-stain electron microscopy of these protein samples would also give some indication as to the oligomeric state of the proteins and the possibility that these samples are aggregated.

The stopped flow kinetics experiments on RsbR indicate bi-exponential dissociation kinetics of O₂ from the protein in agreement with other heme-based sensors.

Statistics

The authors do not perform any statistical analyses of the data presented. This is reasonable given the data presented. All biochemical experiments are presented as replicates with standard deviations.

Conclusions

I have some serious concerns about the conclusions drawn about the formation of stressosome complexes from the proteins and the autophosphorylation of RsbT. The use of MBP/GST fusion proteins and in vitro assembly is likely an inappropriate strategy for the formation of stressosome complexes. RsbR is known to self-associate in the absence of RsbS and these complexes are particularly stable and recalcitrant to the addition of additional components. The use of DLS to identify the formation of complexes does not provide robust evidence as to the size of these and I would suggest simple negative stain EM, and gel filtration to provide firm evidence as to the formation of complexes.

Suggested improvements

In its current form I would not recommend this manuscript for publication.

The conclusions drawn about the autophosphorylation of RsbT require significantly more corroborative evidence. Non-denaturing PAGE and mass spectrometry could be used to identify phosphorylation states and the sites of phosphorylation of this protein and to give alternative evidence.

The production of the proteins as fusions and in vitro mixing will no doubt affect complex formation. I would recommend the production of RsbR and RsbS in a dual expression system to produce stressosomes in vivo and assess their formation by size-exclusion gel-filtration and negative stain electron microscopy. It would be inappropriate to move to cryo-EM experiments at this stage without strong evidence that you are able to produce homogeneous stressosome complexes.

References

The manuscript is appropriately referenced.

Clarity and context

This is a clearly presented manuscript that is well written. I would question the way that the *B. subtilis* stressosome is used to exemplify the *Vibrio* system in figure 1 and throughout the manuscript given the clear differences between the two systems and the lack of a sigma B system in

gram negative bacteria. The authors make no reference to the downstream components of the *Vibrio* stressosome system.

Reviewer #2 (Remarks to the Author):

This paper first reports that the sensor protein, RsbR, is the heme-bound protein and that axial ligand (O₂, NO and CO) binding (or heme-redox state) regulates kinase activities. The results are interesting. However, it is very difficult to follow the whole story. I have several concerns that need to be addressed in order to further strengthen this interesting paper.

This paper did not discuss the molecular mechanism of phosphorylation and phosphotransfer at/between RsbR and RsbS. Identifications of amino acid residues at the phosphorylation sites of RsbR and RsbS with site-directed mutagenesis are encouraged. If authors should properly use site-directed inactive mutants, more valuable information could be obtained, leading to further unraveling of the tangled results in Fig. 3 and Table S1. Mutations at the heme-bound domain would also provide interesting information regarding O₂ binding and the stability of the Fe(II)-O₂ complex.

Table S1: According to Fig. 1, autophosphorylation starts from RsbT, then, RsbR is phosphorylated and finally RsbS is phosphorylated. But, the order in Table S1 is RsbR - RsbS - RsbT. Description in Table S1 is very confusing for readers to grasp the story.

The list of PCR primers should be moved to Supplementary Information to make this paper readable.

Fig. 2 (a): Amino acid residues (proximal histidine, distal tyrosine and serine residues) with numbers involved in the catalysis should be attached on the structure in order to make this interesting paper improved and readable. Tables in Figs. S1 and S11 in Supplementary Information already described those by arrows or highlighted.

Fig. 2 (b): Five spectra all together in one figure make the figure appear complicated. I would suggest the authors to make two separate Figures; for example, one with Fe(II) and Fe(II)-O₂ (inactive and active forms) and other with Fe(II)-NO, Fe(II)-CO and Fe(III). Incorporation of wavelengths of Soret and visible peaks into each spectrum would be beneficial for readers to grasp the heme spectral changes. A new table describing wavelengths of all complexes (rather than written in figure legends) would also be convenient for readers. If the authors should wish to show all species in one figure together, the figure might be incorporated into the Supplementary Information.

Fig. S7: Wavelengths at the Soret and visible regions should be incorporated.

Refs. 23 and 36 are duplicated.

Reviewer #3 (Remarks to the Author):

This work comprises a characterization of Rsb proteins from the Gram-negative bacterium *V. brasiliensis*. These have been identified a decade ago and are homologous to the proteins constituting the well-characterised "stressosome" from the Gram-positive bacterium *B. subtilis*. From sequence alignments, the authors noticed that, different from the *B. subtilis* RsbR protein, which contains a non-heme globin domain, the corresponding domain in *Vibrio* should contain heme. Upon expression and purification of RsbR this indeed turned out to be the case and the authors went on to characterize the effect of complex formation and ligation of RsbR on phosphorylation of complexes comprising also RsbT and RsbS. One major finding that the authors claim is that FeIII and FeII-O₂ complexes of RsbR inhibit RsbS phosphorylation by RsbT, whereas other RsbR complexes do not. The authors conclude that the *V. brasiliensis* stressosome is O₂-regulated. This work would be of considerable interest for a community encompassing both, the stress-response and the heme-based sensor field; many experiments appear solidly performed using classical biochemical techniques (yet some clarifications are required). However I feel the crucial data on the specific inhibition of RsbS phosphorylation are not convincingly presented (see below). Furthermore, some additional data, including in particular a determination of the oxygen affinity of RsbR, must be included in the manuscript. Provided these points would be convincingly addressed, as well as several other points below, I feel this work merits to be published in *Nat. Comm.*

1. The crucial data on the dependence on ligation state of RsbR of phosphorylation of RsbS are collected in Fig. 3b and Table S1. In Fig. 3B the data on FeII, FeII-CO and FeII-NO have large error bars (and a rate of 0 falls almost within experimental error for FeII-CO). The points for the FeIII and FeII-O₂ complexes likewise should have similar error bars, but they are shown as exactly 0 at all times and tabulated as "not detected". For this crucial point to be convincing, the real data and error bars must be shown.
2. On p.5, top paragraph, the O₂ dissociation rate is discussed as if it were the same as the O₂ affinity. This is not the case. If the authors believe it acts as an O₂ sensor, the O₂ affinity itself is highly relevant and titration curves determining such affinities should be shown. Like other O₂ sensors such as FixL, one would expect the affinity to be relatively low so that it is sensible to changes in O₂ partial pressure in the physiological range.
3. p. 4. It is stated that autoxidation occurs with similar rate as myoglobin (where it is 1/day or so), but the rate is not given, and Fig. S7 only shows that it is substantially longer than 2h. A more precise value should be indicated. In the captions of Fig S7, "the beginning" of what exactly?
4. p.3 "The rates....(Figure S5)". First, this is odd phrasing, I think "occurred in a time-dependent manner and" should be deleted. Second, RsbT phosphorylation does not depend on ligation state in Fig. S5b.
5. p.3: "RsbR FeII-NO and FeII-CO yielded intermediate kinase activity... (Table S1)": "intermediate" between what and what? In the plot they seem to be highest.
6. p.4 first phrase: "even though inclusion of RsbR is required..." suggests a contrast that I fail to see. Do the authors mean that in the binary RsbS/RsbT complex no phosphorylation of RsbS is observed (I do not think these data are shown anywhere in the manuscript) and that in the three-component mixture only in the FeII, FeII-CO and FeII-NO complex significant phosphorylation of RsbS is observed? Please clarify.
7. The way the various heme-bound complexes were generated should be described.

8. Fig. 2a: Which template was used for the homology modelling?
9. The procedure to extract rates and their error bars from curves as in Fig. 3 should be described.
10. Fig. S8, panels b and c are missing.
11. Fig. S9. A biexponential fit is better than a monoexponential one, but not that much, especially on the 10 ms timescale, as indicated by the fit residuals. Would a three-exponential fit be better, or would, for that matter, an intrinsic, heterogeneous multiexponential behaviour be an adequate description of the oxygen dissociation kinetics? Please comment.
12. All experiments appear to be performed with stoichiometric ratios for the complexes. Is there evidence that the stressosome stoichiometry should be 1:1:1?

Response to Reviewers

Reviewer #1

“Figure 1 is a very clear depiction of the stressosome, although in the case of *Bacillus subtilis* there is no evidence that RsbT is phosphorylated. There is no published experimental evidence to suggest that the RsbT/SpoIIAB type protein kinases are autophosphorylated.”

- While previous work has shown that SpoIIAB kinases can transfer the terminal phosphate directly to SpoIIAA, there are other residues on *Vb* RsbT that could potentially be phosphorylated. *V. brasiliensis* RsbT does not have high sequence identity to *B. subtilis* SpoIIAB (18% amino acid sequence identity; 38% sequence similarity), with a number of mutations within the putative ATP binding pocket and a three amino acid deletion that includes an arginine used to stabilize ATP binding in *B. stearothermophilus* SpoIIAB (Masuda, *et al. J. Mol. Biol.* **2004**, *340*, 941-956). Therefore, even though autophosphorylated SpoIIAB proteins previously have not been observed, it is not surprising that *Vb* RsbT exhibits different characteristics. Furthermore, autophosphorylated kinases sometimes are not observed experimentally due to the instability of phosphorylated intermediate. It has previously been shown that the autophosphorylated state of a number of bacterial kinases can not be observed using ³²P-ATP for visualization, but usage of ATP-γ-S (resulting in thiophosphate transfer) results in robust detection of autophosphorylation (Carlson, *et al. Anal. Biochem.* **2010**, *397*, 139-43). The observation of previously undetectable thiophosphorylated kinases is due to the increased stability of thiophosphorylated amino acids; thiophosphates are less electrophilic, decreasing the rate of hydrolysis (Carlson, *et al. Anal. Biochem.* **2010**, *397*, 139-43; Lasker, *et al. Protein Sci.* **1999**, *8*, 2177-85). Therefore, our usage of ATP-γ-S as the kinase substrate was fortuitous in its ability to stabilize phosphorylated RsbT, allowing us to observe kinase autophosphorylation. The legend of Figure 1 has been updated to make clear that the stressosome depiction is a proposed model of the *V. brasiliensis* system, not the *B. subtilis* system.

“The authors produce the RsbR/S/T proteins as fusions with MBP/GST and purify these by affinity chromatography. The use of fusion proteins is justified as a way of enhancing protein stability, although this can affect protein/protein interactions in protein complexes.”

- RsbR/RsbS/RsbT proteins that have been cleaved from their tags have been purified and characterized to address concerns regarding the influence of the tags and aggregation. Based on additional DLS and AGF data, the tag-cleaved proteins are less stable than the

tagged Rsb proteins, and assemble/aggregate in solution to form extremely large complexes (Supplementary Figures 10 and 11).

“The heme-containing RsbR protein is characterised by UV/visible spectroscopy and the authors identify absorption peaks consistent with the presence of heme in the protein. The spectrum shown in Figure 2b lacks any indication of values, or units on the y-axis. I appreciate that for UV/visible spectroscopy these units are arbitrary, but it would be instructive to see the absolute values recorded.”

- Units have been added to the y-axis.

“The phosphorylation kinetics experiments presented suggest that RsbT is competent to phosphorylate RsbR and RsbT in the presence of RsbR in various ligation states. These results do not show particularly strong effects for different ligation states in the case of RsbR, while ligation with oxygen and the Fe^{III} states appear to suppress RsbS phosphorylation. Given the size of the errors on these data it is not clear that the effects of the different ligation states of RsbR and RsbT are significantly different.”

- The rates of phospho-transfer to RsbR are significantly different and outside of the error bars when the Fe^{II} unligated (active complex) and Fe^{II}-O₂ (basal activity complex) curves (Fig. 3A) and rates (Table S2) are compared. The intermediate rates of phospho-transfer to RsbR in the Fe^{II}-NO and Fe^{II}-CO complexes make the difference less obvious, as those time points fall in between the Fe^{II} and Fe^{II}-O₂ curves. Figure 3 has been updated to show only the Fe^{II} and Fe^{II}-O₂ data for easier comparison, with data for all complexes moved to Supplementary Figure 7.

“I have significant concerns about the identification of autophosphorylation of RsbT. There are no reports of this type of kinase exhibiting such behaviour and this observation requires strong corroborative evidence to rule out potential false positives. It would be instructive to see a non-denaturing PAGE gel with RsbT incubated with ATP to see if this protein is autophosphorylated would provide strong evidence to show that this effect is indeed real. Mass spectrometry could also be used to identify the position on which RsbT is autophosphorylated.”

- As described in more detail above, phosphorylated kinases often are quite prone to hydrolysis, making the autophosphorylated kinases challenging to detect. In addition, *V. brasiliensis* RsbT may likely be utilizing a different mechanism than the classis Gram-positive SpoIIAB kinases are thought to use, as it is from a Gram-negative bacterium and does not maintain high sequence similarity. Incubation of RsbT or the RsbR/S/T complex under kinase assay conditions without ATP-γ-S does not lead to any observable signal above background. In addition, heat denaturation of RsbT (either alone or in the RsbR/S/T complex) prior to addition of ATP-γ-S also does not result in an observable signal and mutation of an aspartic acid proposed to be involved in phosphoryl transfer results in inactive RsbT (Supplementary Figure 4). To further investigate RsbT autophosphorylation, we have performed a mass spectrometry-based proteomic analysis of thiophosphorylated RsbT and identified phosphorylation sites in an α-helix that is involved in protein-protein interactions with σ^F in *B. stearothermophilus* SpoIIAB (Campbell, *et al. Cell* **2002**, *108*, 795-807; manuscript text, Supplementary Table 4, Supplementary Figure 9).

“The Dynamic light scattering experiments used to assess the formation of stressosome complexes shown in Figure S7 appear to show that RsbR and RsbS form large aggregated of around 30 nm diameter and the R/S/T mixture is much smaller at around 12 nm. These results give rise to serious concerns about the state of these proteins in solution. It would be highly instructive to attempt gel-filtration of the individual proteins and protein mixtures to assess whether the proteins are indeed forming aggregates. Simple negative-stain electron microscopy of these protein samples would also give some indication as to the oligomeric state of the proteins and the possibility that these samples are aggregated.”

- RsbR/RsbS/RsbT proteins that have been cleaved from their tags have been purified and characterized to address concerns regarding the influence of the tags and aggregation. The phosphorylation reaction with tag-cleaved RsbR/S/T previously had been subjected to preliminary testing and was found to proceed with similar activity and kinetics as the tagged proteins. However, the tag-cleaved proteins were not as stable for the length of time required to set up and run the kinase assays and began aggregating by the end of the 120 min. reaction, which is why tagged proteins were used for all kinase assays described in the text.
- DLS and analytical gel filtration (AGF) experiments were conducted for tag-cleaved RsbR, RsbS, RsbT, and the RsbR/S/T complex (Supplementary Figures 10 and 11). From these studies, the tag-cleaved RsbR/S/T complex was observed to have a larger diameter (30.2 nm) than the tagged RsbR/S/T complex (15.8 nm), suggesting that the tags may have precluded formation of a full stressosome complex. However, DLS and AGF of the individual proteins also showed larger complexes/aggregates for the untagged proteins, as compared with the tagged proteins. This may be due to the instability of the tag-cleaved proteins and initial formation of aggregates in solution. In addition, formation of the smaller complexes with the tagged proteins clearly did not prohibit kinase activity (Figure 3, Supplementary Figure 7). For both tagged and untagged proteins, larger aggregates were observed for RsbR and RsbS, suggesting that the STAS domains form protein-protein interactions that lead to large oligomer formation. Similar results were observed by AGF, with all of the proteins forming large complexes and only smaller peaks at lower molecular weights (Supplementary Figure 11). The greater percentage of proteins in high molecular weight complexes likely is due to the length of time required for AGF analysis, as compared to DLS analysis; as the tag-cleaved proteins sit, a greater percentage begins forming large complexes or aggregates (as described above). Even so, co-elution of the proteins can be observed when comparing the absorbance at 280 nm and 416 nm (maximal absorbance of RsbR heme in the Fe^{II}-O₂ ligation state). For RsbR alone, the 280 nm:416 nm ratio is 1.2:1, while for RsbR/S/T complex the ration is 1.7:1 (Supplementary Figure 11), which shows that heme-free proteins (RsbS and RsbT) are co-eluting with RsbR, leading to the increase in absorbance at 280 nm and substantiating the formation of a complex.

“In its current form I would not recommend this manuscript for publication. The conclusions drawn about the autophosphorylation of RsbT require significantly more corroborative evidence. Non-denaturing PAGE and mass spectrometry could be used to identify phosphorylation states and the sites of phosphorylation of this protein and to give alternative evidence.”

- A discussion of autophosphorylation of RsbT and the additional mass spectrometry-based proteomic experiments undertaken to address the concerns are described above.

“The production of the proteins as fusions and in vitro mixing will no doubt affect complex formation. I would recommend the production of RsbR and RsbS in a dual expression system to produce stressosomes in vivo and assess their formation by size-exclusion gel-filtration and negative stain electron microscopy. It would be inappropriate to move to cryo-EM experiments at this stage without strong evidence that you are able to produce homogeneous stressosome complexes.”

- Dual expression systems for RsbR and RsbS were previously attempted, but did not lead to any observable protein expression under the wide range of expression conditions that were tested. Similarly, expression of individual Rsb proteins without a solubility tag resulted in extremely poor expression yields and protein that aggregated during the multi-column purification process. As described above, RsbR/S/T proteins were cleaved from their tags and assessed by DLS and AGF to address the concerns regarding the solubility tags.

“This is a clearly presented manuscript that is well written. I would question the way that the *B. subtilis* stressosome is used to exemplify the *Vibrio* system in figure 1 and throughout the manuscript given the clear differences between the two systems and the lack of a sigma B system in gram negative bacteria. The authors make no reference to the downstream components of the *Vibrio* stressosome system.”

- Comments about possible downstream components of *V. brasiliensis* Rsb system and differences as compared to the *B. subtilis* stressosome have been added to the text.

Reviewer #2

“This paper did not discuss the molecular mechanism of phosphorylation and phosphotransfer at/between RsbR and RsbS. Identifications of amino acid residues at the phosphorylation sites of RsbR and RsbS with site-directed mutagenesis are encouraged. If authors should properly use site-directed inactive mutants, more valuable information could be obtained, leading to further unraveling of the tangled results in Fig. 3 and Table S1. Mutations at the heme-bound domain would also provide interesting information regarding O₂ binding and the stability of the Fe(II)-O₂ complex.”

- Given the low sequence similarity between *V. brasiliensis* stressosome proteins and previously characterized *Bacillus* proteins (RsbR = 26%; RsbS = 28%; RsbT = 18%), and that a number of *Bacillus* stressosome protein phosphorylation sites are residues that can't be phosphorylated in *Vb* stressosome proteins, a mass spectrometry-based proteomics approach was used to identify phosphorylation sites in RsbR/S/T (manuscript text, Supplementary Table 4, Supplementary Figure 9). Amino acids that are phosphorylated were identified in RsbR and RsbT. Unfortunately we were unable to identify phosphorylated peptides of RsbS, possibly due to the lower levels of RsbS phosphorylation in the reaction (Supplementary Figure 4b).

“Table S1: According to Fig. 1, autophosphorylation starts from RsbT, then, RsbR is phosphorylated and finally RsbS is phosphorylated. But, the order in Table S1 is RsbR - RsbS - RsbT. Description in Table S1 is very confusing for readers to grasp the story.”

- Table S1 has been modified to address the critique.

“The list of PCR primers should be moved to Supplementary Information to make this paper readable.”

- The PCR primer list has been moved to the supporting information, as suggested.

“Fig. 2 (a): Amino acid residues (proximal histidine, distal tyrosine and serine residues) with numbers involved in the catalysis should be attached on the structure in order to make this interesting paper improved and readable. Tables in Figs. S1 and S11 in Supplementary Information already described those by arrows or highlighted.”

- Amino acids labels have been added to Figure 2A.

“Fig. 2 (b): Five spectra all together in one figure make the figure appear complicated. I would suggest the authors to make two separate Figures; for example, one with Fe(II) and Fe(II)-O₂ (inactive and active forms) and other with Fe(II)-NO, Fe(II)-CO and Fe(III). Incorporation of wavelengths of Soret and visible peaks into each spectrum would be beneficial for readers to grasp the heme spectral changes. A new table describing wavelengths of all complexes (rather than written in figure legends) would also be convenient for readers. If the authors should wish to show all species in one figure together, the figure might be incorporated into the Supplementary Information.”

- Figure 2b has been updated to improve the figure. In addition, a table listing wavelengths of the various complexes has been added to the supplementary information.

“Fig. S7: Wavelengths at the Soret and visible regions should be incorporated.”

- Wavelengths have been incorporated as requested.

“Refs. 23 and 36 are duplicated.”

- The referencing error has been corrected.

Reviewer #3: Response to Reviewer

“1. The crucial data on the dependence on ligation state of RsbR of phosphorylation of RsbS are collected in Fig. 3b and Table S1. In Fig. 3B the data on FeII, FeII-CO and FeII-NO have large error bars (and a rate of 0 falls almost within experimental error for FeII-CO). The points for the FeIII and FeII-O₂ complexes likewise should have similar error bars, but they are shown as exactly 0 at all times and tabulated as "not detected". For this crucial point to be convincing, the real data and error bars must be shown.

2.”

- Figure 3 and the discussion have been modified to improve the presentation and address the concern. As all band intensities are normalized to the largest signal for each protein (RsbR, RsbS, or RsbT) on each Western blot to control for any blot to blot differences in staining, the lack of signal above background for RsbS under Fe^{III} and Fe^{II}-O₂ conditions leads to random changes in band intensity with large errors.

“On p.5, top paragraph, the O₂ dissociation rate is discussed as if it were the same as the O₂ affinity. This is not the case. If the authors believe it acts as an O₂ sensor, the O₂ affinity itself is highly relevant and titration curves determining such affinities should be shown. Like other O₂ sensors such as FixL, one would expect the affinity to be relatively low so that it is sensible to changes in O₂ partial pressure in the physiological range.

3. p. 4.”

- Oxygen association rates and binding affinities have been measured, and added to Table 1 and the discussion.

“It is stated that autoxidation occurs with similar rate as myoglobin (where it is 1/day or so), but the rate is not given, and Fig. S7 only shows that it is substantially longer than 2h. A more precise value should be indicated.”

- The autoxidation rate has been reported in the text.

“In the captions of Fig S7, "the beginning" of what exactly?”

- The figure legend was updated to clarify the data.

“4. p.3 "The rates....(Figure S5)". First, this is odd phrasing, I think "occurred in a time-dependent manner and" should be deleted. Second, RsbT phosphorylation does not depend on ligation state in Fig. S5b.”

- The text has been updated to reflect the critique.

“5. p.3: "RsbR Fe^{II}-NO and Fe^{II}-CO yielded intermediate kinase activity... (Table S1)": "intermediate" between what and what? In the plot they seem to be highest.”

- RsbR Fe^{II}-CO or Fe^{II}-NO yield intermediate rates of phosphorylation of RsbR and RsbS, but the greatest rate of RsbT phosphorylation. This is likely because RsbT is more rapidly transferring the phosphate to RsbR and RsbS when RsbR Fe^{II} is included. The text has been updated to clarify this point.

“6. p.4 first phrase: "even though inclusion of RsbR is required..." suggests a contrast that I fail to see. Do the authors mean that in the binary RsbS/RsbT complex no phosphorylation of RsbS is observed (I do not think these data are shown anywhere in the manuscript) and that in the three-component mixture only in the Fe^{II}, Fe^{II}-CO and Fe^{II}-NO complex significant phosphorylation of RsbS is observed? Please clarify.”

- Figure S4C shows a representative reaction of RsbS/RsbT run under the same conditions as the RsbR/RsbS/RsbT reactions. RsbT autophosphorylation can be clearly observed, but there is no detectable phosphorylation of RsbS. In the three-component mixture, RsbS phosphorylation never rises above background or shows any time dependence when Fe^{II}-O₂ or Fe^{III} RsbR is included in the mixture. In contrast, inclusion of Fe^{II}, Fe^{II}-CO, or Fe^{II}-NO RsbR leads to readily observable phosphorylation of RsbS.

“7. The way the various heme-bound complexes were generated should be described.”

- The appropriate information was added to the methods section.

“8. Fig. 2a: Which template was used for the homology modelling?”

- The PDBID for the structure used for homology modeling was added to the figure legend.

“9. The procedure to extract rates and their error bars from curves as in Fig. 3 should be described.”

- The procedural information has been added to the Table S1 figure legend.

“10. Fig. S8, panels b and c are missing.”

- The mistake in the text has been corrected.

“11. Fig. S9. A biexponential fit is better than a monoexponential one, but not that much, especially on the 10 ms timescale, as indicated by the fit residuals. Would a three-exponential fit be better, or would, for that matter, an intrinsic, heterogeneous multiexponential behaviour be an adequate description of the oxygen dissociation kinetics? Please comment.”

- The residuals in the first ~10-15 ms of the stopped flow reaction are due to disturbances of mixing in the instrument and are always visible for fast oxygen dissociation reactions that require many data points at short wavelengths (for an example with different heme proteins, see Burns, *et al. Mol. BioSyst.* **2014**, *10*, 2823-2826). A three-exponential fit does not improve the fitting at times longer than 10ms, suggesting that two exponentials can

adequately describe the behavior. Previous studies that identified open and closed conformations of the *Bacillus subtilis* globin coupled sensor heme pocket that are correlated with different O₂ dissociation rates (Ohta, *et al. J. Am. Chem. Soc.* **2004**, *126*, 15000-15001; Zhang, *et al. Biophys. J.* **2005**, *88*, 2801-2804), and no other data has been reported to suggest that incorporation of additional fitting parameters is justified. Therefore, a biexponential fit is currently supported as the best option.

“12. All experiments appear to be performed with stoichiometric ratios for the complexes. Is there evidence that the stressosome stoichiometry should be 1:1:1?”

- The *Bacillus subtilis* stressosome previously was found to have a 2:1:1 RsbR:S:T stoichiometry (Marles-Wright, *et al. Science*, **2008**, *322*, 92-96), with RsbR proteins forming dimers in the full complex. However, when other ratios, including 2:1:1, initially were tested, the level of phosphorylation and time dependence of the reaction did not change so we maintained the 1:1:1 stoichiometry for all reactions described in the manuscript because it resulted in the clearly detectable phosphorylation of all three proteins. Doubling the concentration of RsbR (to yield a 2:1:1 ratio) resulted in Western blots where either the RsbR signal was outside the linear range at later time points or the RsbS signal was barely observable.

Reviewers' comments:

Reviewer #1 (Remarks to the Author):

I would like to thank the authors for their consideration of the comments on their manuscript. They have addressed these comments appropriately and I would now recommend publication.

Reviewer #2 (Remarks to the Author):

[1] I am curious if the present work is associated with the two-component signal transduction system. Asp residue(s) of the cognate response regulator (second component) is phosphorylated in response to the activation (autophosphorylation at His) of heme-based oxygen sensors (FixL, DevS, DosT, AfGCHK) (first component). The present work looks similar to the system. Please discuss this point somewhere in the main text or Supplementary Information.

[2] From Fig. 1, I understand that

No stress

1. T is first autophosphorylated.
2. The phospho group of T transfers to R. Then, R is phosphorylated at D87. (Similar to the bacterial two-component signal transduction system?)

With stress

1. T is first autophosphorylated.

2. The phospho group of T transfers to R. Then, R is phosphorylated at D87.
3. R is further multi-phosphorylated.
4. S is phosphorylated (by phosphor-transfer reaction from phosphorylated T or phosphorylated R?).
5. Phosphorylation of S results in dissociation of phosphorylated T from the stressosome core.
6. Dissociated/isolated/free phosphorylated T regulates or activates downstream gene transcription.

The original and revised manuscripts do not clearly explain those successive reactions in this way. Am I wrong? If I am wrong, the manuscript does not clearly explain readers about the successive reactions.

Then, Fig. 3a describes phosphorylation of R is regulated by its own heme iron coordination forms of R, Fe(II) and Fe(II)-O₂ forms in the mixture T/R/S. Phosphorylation of S is regulated by the heme iron coordination forms of R, Fe(II) and Fe(II)-O₂ forms (Fig. 3b) (the authors say Fe(II)-O₂ form results are unclear). Then, what is the phosphorylation of T (Fig. 3c)? Does Fig. 3b indicate that first autophosphorylation of T is not regulated by the heme iron coordination of R, Fe(II) and Fe(II)-O₂? This is not surprising, but nonsense.

Supp. Fig. 5 describes phosphorylation of R is regulated by its own heme iron coordination forms of R, Fe(II) and Fe(II)-O₂ forms in the mixture T/R. In contrast, first autophosphorylation of T is not regulated by the heme iron coordination of R, Fe(II) and Fe(II)-O₂. This is also not surprising, but nonsense.

Supp. Fig. 7 describes the same reactions as Fig. 3 in the presence of various heme iron coordination forms, Fe(III), Fe(II)-O₂, Fe(II)-CO, Fe(II)-NO and Fe(II) unligated.

[3] Please see J. Biol. Chem. 39, 27702 (2013). This mini-review article summarizes structure and function relationships of the globin-coupled oxygen sensors and other heme-based oxygen sensors. HemAC-Lm (PNAS 42, 16790 (2013); BBA 1844, 615 (2014), ABB 579, 85 (2015)) will be added as an important GCS.

[4] Refs. 23 and 44 are duplicated.

[5] Legend for Table 1. d, Ref. 37

[6] The stress-induced consecutive reactions described in Fig. 1 look too small, still difficult for general readers to understand the whole story what the authors have done and wish to emphasize in this work. Please make a new cartoon (expanded from Fig. 1) in Supplementary Information (or in the main text) which describes the successive (or revolving) reactions and correlations between RsbR, RsbT, and RsbS occurring under the stress conditions. The cartoon should be comprehensive and include heme redox and ligand (oxygen)-free, -bound states, phosphorylation residue(s), heme proximal ligand, Tyr residue stabilizing the heme Fe(II)-O₂ complex, protein-protein interaction, protein binding affinity, and putative down stream gene transcription (aerotaxis, CheY?) and so on. Please note that a clear and comprehensive cartoon will be often used for lectures (by other people) to demonstrate the importance of this work.

Reviewer #3 (Remarks to the Author):

My concerns have been convincingly addressed. Following the reply of the authors, I have two minor suggestions for improvement they may want to incorporate:

- referring to my first comment, I understand now that each curve is actually normalized on itself, and that the relative amplitude of the two curves in Fig. 3b is arbitrary. For clarity the red dotted curve may therefore might also be plotted with lower amplitude.
- referring to the reply to my comment 12, I feel that this information clarifies the strategy and should therefore also be given in the paper itself.

Response to Reviewers

Reviewer #1:

I would like to thank the authors for their consideration of the comments on their manuscript. They have addressed these comments appropriately and I would now recommend publication.

- Thank you for your comments.

Reviewer #2:

[1] I am curious if the present work is associated with the two-component signal transduction system. Asp residue(s) of the cognate response regulator (second component) is phosphorylated in response to the activation (autophosphorylation at His) of heme-based oxygen sensors (FixL, DevS, DosT, AfGCHK) (first component). The present work looks similar to the system. Please discuss this point somewhere in the main text or Supplementary Information.

- RsbT is not a histidine kinase and we do not observe any phosphorylated histidine or aspartic acid residues. A comparison of stressosome signaling and two-component signaling has been added to the introduction.

[2] From Fig. 1, I understand that

No stress

1. T is first autophosphorylated.
2. The phospho group of T transfers to R. Then, R is phosphorylated at D87.

(Similar to the bacterial two-component signal transduction system?)

With stress

1. T is first autophosphorylated.
2. The phospho group of T transfers to R. Then, R is phosphorylated at D87.
3. R is further multi-phosphorylated.
4. S is phosphorylated (by phosphor-transfer reaction from phosphorylated T or phosphorylated R?).
5. Phosphorylation of S results in dissociation of phosphorylated T from the stressosome core.
6. Dissociated/isolated/free phosphorylated T regulates or activates downstream gene transcription.

The original and revised manuscripts do not clearly explain those successive reactions in this way. Am I wrong? If I am wrong, the manuscript does not clearly explain readers about the successive reactions.

- Figure 1 has been updated to represent the previously described stressosome from *B. subtilis*. A full description of the signaling pathway described in the manuscript has been added as Supplementary Figure 15. Residue D87 described in the text is on kinase RsbT, not sensor RsbR. A D87N RsbT mutant was generated and characterized based on previous studies of the *B. subtilis* RsbT (Kang, *et al. Mol. Microbio.* **1998**, *30*, 189-196), *V. brasiliensis* D87N RsbT was inactive. Based on our studies and previous work on the *B. subtilis* stressosome, it appears that in the *V. brasiliensis* stressosome, RsbT is first autophosphorylated, then RsbT phosphorylates RsbR. High levels of RsbR phosphorylation lead to RsbT phosphorylating RsbS (we do not observe any phosphorylation of RsbS in a RsbR/RsbS mixture and *B. subtilis* RsbT directly phosphorylates RsbS (Kang, *et al. Mol. Microbio.* **1998**, *30*, 189-196)). Following phosphorylation of RsbR and RsbS, RsbT should be able to activate downstream signaling pathways. The text and figure legends have been update to improve clarity.

Then, Fig. 3a describes phosphorylation of R is regulated by its own heme iron coordination forms of R, Fe(II) and Fe(II)-O₂ forms in the mixture T/R/S. Phosphorylation of S is regulated by the heme iron coordination forms of R, Fe(II) and Fe(II)-O₂ forms (Fig. 3b) (the authors say Fe(II)-O₂ form results are unclear). Then, what is the phosphorylation of T (Fig. 3c)? Does Fig. 3b indicate that first autophosphorylation of T is not regulated by the heme iron coordination of R, Fe(II) and Fe(II)-O₂? This is not surprising, but nonsense.

- We cannot measure the rate of RsbT autophosphorylation in the presence of RsbR and RsbS. Instead, we measure an observed level of RsbT phosphorylation, which is dictated by the competing reactions of autophosphorylation and phosphoryl-transfer to RsbR/RsbS. The overall activity of RsbT is clearly increased for RsbR Fe^{II}, as compared to RsbR Fe^{II}-O₂, suggesting that the autophosphorylation rate also is increased. The text has been updated to clarify this point.

Supp. Fig. 5 describes phosphorylation of R is regulated by its own heme iron coordination forms of R, Fe(II) and Fe(II)-O₂ forms in the mixture T/R. In contrast, first autophosphorylation of T is not regulated by the heme iron coordination of R, Fe(II) and Fe(II)-O₂. This is also not surprising, but nonsense.

- As described for the previous comment, a true rate of RsbT autophosphorylation cannot be measured due to the phosphoryl-transfer reaction that depletes phospho-RsbT and forms phospho-RsbR.

Supp. Fig. 7 describes the same reactions as Fig. 3 in the presence of various heme iron coordination forms, Fe(III), Fe(II)-O₂, Fe(II)-CO, Fe(II)-NO and Fe(II) unligated.

- Figure 3 only shows Fe^{II} and Fe^{II}-O₂ rates for more facile comparison. Supp. Fig. 7 compares all of the ligation/oxidation states that were tested.

[3] Please see J. Biol. Chem. 39, 27702 (2013). This mini-review article summarizes structure and function relationships of the globin-coupled oxygen sensors and other heme-based oxygen sensors. HemAC-Lm (PNAS 42, 16790 (2013); BBA 1844, 615 (2014), ABB 579, 85 (2015)) will be added as an important GCS.

- All of the requested references have been added.

[4] Refs. 23 and 44 are duplicated.

- The duplicate references have been removed.

[5] Legend for Table 1. d, Ref. 37

- The error has been corrected.

[6] The stress-induced consecutive reactions described in Fig. 1 look too small, still difficult for general readers to understand the whole story what the authors have done and wish to emphasize in this work. Please make a new cartoon (expanded from Fig. 1) in Supplementary Information (or in the main text) which describes the successive (or revolving) reactions and correlations between RsbR, RsbT, and RsbS occurring under the stress conditions. The cartoon should be comprehensive and include heme redox and ligand (oxygen)-free, -bound states, phosphorylation residue(s), heme proximal ligand, Tyr residue stabilizing the heme Fe(II)-O₂ complex, protein-protein interaction, protein binding affinity, and putative down stream gene transcription (aerotaxis, CheY?) and so on. Please note that a clear and comprehensive cartoon will be often used for lectures (by other people) to demonstrate the importance of this work.

- A comprehensive figure depicting all of the insights gained from this work has been added to the Supplementary Information (Supp. Fig. 15).

Reviewer #3:

My concerns have been convincingly addressed. Following the reply of the authors, I have two minor suggestions for improvement they may want to incorporate:

[1] Referring to my first comment, I understand now that each curve is actually normalized on itself, and that the relative amplitude of the two curves in Fig. 3b is arbitrary. For clarity the red dotted curve may therefore also be plotted with lower amplitude.

- The graph has been replotted as suggested, using the average intensity for RsbS in the Fe^{II} reactions to yield a lower amplitude.

[2] Referring to the reply to my comment 12, I feel that this information clarifies the strategy and should therefore also be given in the paper itself.

- The information about RsbR/S/T ratios has been added to the methods section.